# FRET-enhanced photostability allows improved single-molecule tracking of proteins and protein complexes in live mammalian cells

Srinjan Basu[1], Lisa-Maria Needham[2], David Lando [1], Edward J.R. Taylor[1], Kai J. Wohlfahrt[1], Devina Shah[1], Wayne Boucher[1], Yi Lei Tan[1], Lawrence E. Bates [1], Olga Tkachenko[1], Julie Cramard[3], B. Christoffer Lagerholm[4], Christian Eggeling[4], Brian Hendrich [1,3], Dave Klenerman[2], Steven F. Lee[2] & Ernest D. Laue [1]

A major challenge in single-molecule imaging is tracking the dynamics of proteins or complexes for long periods of time in the dense environments found in living cells. Here, we introduce the concept of using FRET to enhance the photophysical properties of photomodulatable (PM) fluorophores commonly used in such studies. By developing novel single-molecule FRET pairs, consisting of a PM donor fluorophore (either mEos3.2 or PA-JF$_{549}$) next to a photostable acceptor dye JF$_{646}$, we demonstrate that FRET competes with normal photobleaching kinetic pathways to increase the photostability of both donor fluorophores. This effect was further enhanced using a triplet-state quencher. Our approach allows us to significantly improve single-molecule tracking of chromatin-binding proteins in live mammalian cells. In addition, it provides a novel way to track the localization and dynamics of protein complexes by labeling one protein with the PM donor and its interaction partner with the acceptor dye.

[1] Department of Biochemistry, University of Cambridge, 80 Tennis Court Road, Cambridge CB2 1GA, UK. [2] Department of Chemistry, University of Cambridge, Lensfield Road, Cambridge CB2 1EW, UK. [3] Wellcome Trust – MRC Stem Cell Institute, University of Cambridge, Tennis Court Road, Cambridge CB2 1QR, UK. [4] Medical Research Council Human Immunology Unit and Wolfson Imaging Centre, Weatherall Institute of Molecular Medicine, University of Oxford, Headley Way, Oxford OX3 9DS, UK. These authors contributed equally: Srinjan Basu, Lisa-Maria Needham. Correspondence and requests for materials should be addressed to D.K. (email: dk10012@cam.ac.uk) or to S.F.L. (email: sl591@cam.ac.uk) or to E.D.L. (email: e.d.laue@bioc.cam.ac.uk)

Single-molecule fluorescence imaging approaches have allowed one to study the localization and dynamics of single proteins in live mammalian cells, shedding light on cellular processes such as how proteins bind to chromatin to regulate gene transcription[1–7]. Obtaining long trajectories is extremely informative as it allows the detection of rare events, such as transitions of a single molecule between different types of motion[8]. However, this is often limited by the photostability of fluorophores. Another challenge has been to distinguish whether a protein is moving alone or as part of a particular protein complex. The photon-limited localization precision of two color coincident detection experiments often prevents one from determining whether two proteins really are part of the same complex or simply localized in proximity to each other. Complementation methods have been developed to image single protein complexes within living cells[9–12], but improved approaches are needed to increase the length of time that the molecules can be tracked. For this reason, methods that allow single-molecule tracking for extended periods of time in the densely packed interior of live cells would be very valuable.

Generally, single-molecule tracking requires the point spread functions of individual fluorophores to be spatially separated during the imaging process. In the common case of high fluorophore density, this can be achieved either spatially by under-labeling or temporally as in single-particle tracking (SPT) using photo-activated localization microscopy (sptPALM)[3,6,13]. Up to now, most single-molecule tracking studies have employed conventional organic dyes[4,5,7], due to both their relative brightness and photo-stability compared to fluorescent proteins. Under-labeling approaches are typically employed[4,7], but they only allow imaging of the dynamics of relatively few molecules in each cell. In contrast, temporal control to image single molecules in dense environments, such as the cell nucleus, is greatly facilitated by using the highly controllable photophysical properties of photo-modulatable (PM) fluorophores. PM fluorophores exist in either a fluorescently active (on) or fluorescently inactive (off) state. They can be categorized into distinct classes: photo-activatable fluorophores can be activated from a non-emissive to an emissive fluorescent state[14], and photo-convertible fluorophores can be converted from one emissive state to another[15,16]. Although most PM fluorophores are still currently fluorescent proteins[3] (PM FPs), cell-permeable, photo-activatable dyes have recently been developed[17]. Novel strategies to enhance the photophysical properties, such as photostability, of PM fluorophores are therefore highly desirable.

Here, we introduce the concept of using Förster resonance energy transfer (FRET)[18–22] to regulate the fluorescence properties of PM fluorophores and then use this approach to extend their trajectory lengths for studies of both single proteins and protein complexes in dense environments in live mammalian cells. We place a photostable organic dye in close spatial proximity to a PM donor fluorophore (Fig. 1a) and use FRET to non-radiatively transfer energy from the PM donor fluorophore to the acceptor dye via dipole–dipole coupling. In so doing, we were able to modify the excited-state kinetics of the donor PM fluorophore and selectively tune photophysical properties such as the fluorescence lifetime[23] and photostability—by providing additional energetic pathways for return to the ground state instead of photobleaching. This allowed us to retain the properties of specific PM fluorophores for single-molecule imaging and to exploit a known approach for stabilizing dye molecules to improve the photostability and photon budget of two PM fluorophores, photo-convertible mEos3.2[15] and photo-activatable dye PA-JF$_{549}$[17]. We then used this approach to track proteins and complexes for substantially longer in live mammalian cells, by either labeling a protein with both the PM donor fluorophore and the acceptor dye, or by labeling one protein with the PM donor and its interaction partner with the acceptor dye.

## Results

**mEos3.2–JF$_{646}$ as a novel single-molecule FRET pair**. To investigate whether FRET can be exploited to modulate the properties of existing PM fluorophores, we explored a number of potential donor–acceptor pairs. An optimal FRET pair requires high donor quantum yield, high acceptor extinction coefficient, and significant overlap between the donor emission and acceptor absorption spectra[22]. In initial experiments, we explored a number of potential FRET pairs such as PA-EGFP-mCherry and PS-CFP2-YPet, but these were unsuccessful due to problems of poor acceptor photostability, folding inefficiencies, tendencies to multimerize, high levels of pre-conversion, and limited donor brightness for live-cell single-molecule tracking. These initial experiments led us to focus on mEos3.2[15] as the donor fluorophore with organic dyes such as JF$_{646}$ as the acceptor, which can be covalently tethered to either the HaloTag[24] or SNAP-tag[25,26] proteins (Fig. 1b), providing flexibility for different experiments. mEos3.2 has a high quantum yield (Supplementary Fig. 1a), is monomeric to prevent aggregation in dense protein environments[15], and is widely used for live-cell single-molecule imaging. JF$_{646}$ is photostable, membrane-permeable, has a high extinction coefficient (Supplementary Fig. 1b, c) and there is good spectral overlap between its absorption spectrum and the emission spectrum of mEos3.2 (Supplementary Fig. 1d).

We created a test system by expressing a mEos3.2-HaloTag fusion with a short linker between the two proteins (Supplementary Fig. 2) to ensure that the distance between the donor and acceptor fluorophores was less than 10 nm for efficient FRET (the radii of the mEos3.2 and the HaloTag proteins are approximately 1.5 and 2.0 nm, respectively). To confirm FRET, we measured the fluorescence lifetime of mEos3.2 after photo-conversion, both with and without bound JF$_{646}$. For mEos3.2, we found a mean singlet excited-state lifetime of 3.6 ns (Fig. 1c, Supplementary Table 2), consistent with previously measured values for red FPs[27]. As expected, upon addition of JF$_{646}$, the mean fluorescence lifetime of the photo-converted mEos3.2 donor decreased to 1.8 ns (Fig. 1c). The decrease in excited-state lifetime corresponded to a FRET efficiency of $0.49 \pm 0.04$ (mean ± s.d.) and a mean inter-fluorophore distance of $6 \pm 1$ nm (see Methods). We also collected bulk fluorescence emission spectra and demonstrated a decrease in the donor and an increase in the acceptor fluorescence after taking into account the fluorescence arising from direct excitation of JF$_{646}$ at the donor excitation wavelength (Fig. 1d). To further demonstrate that this was dependent on FRET, we expressed and purified a second mEos3.2-HaloTag fusion protein with no linker between the two proteins, and found that the FRET efficiency increased correspondingly (Supplementary Fig. 2d). Taken together these data confirm that mEos3.2–JF$_{646}$ undergoes FRET after photo-conversion.

**FRET modulation improves the photophysics of mEos3.2**. To explore the photophysical parameters of the photo-convertible mEos3.2–JF$_{646}$ FRET pair for single-molecule fluorescence imaging, we recorded TIRF images of mEos3.2 with and without JF$_{646}$ after immobilizing the relevant fusion proteins on a surface at low spatial density (Fig. 1d, Supplementary Fig. 3). The single-molecule photophysical parameters were then characterized using Hidden Markov Modeling (see Methods). We determined the number of photons detected in a single frame, the total number of emitted photons, the total time mEos3.2 remained in the fluorescent on-state, the on-state time for each photoswitching (blinking) event, and the number of reversible on-state to off-

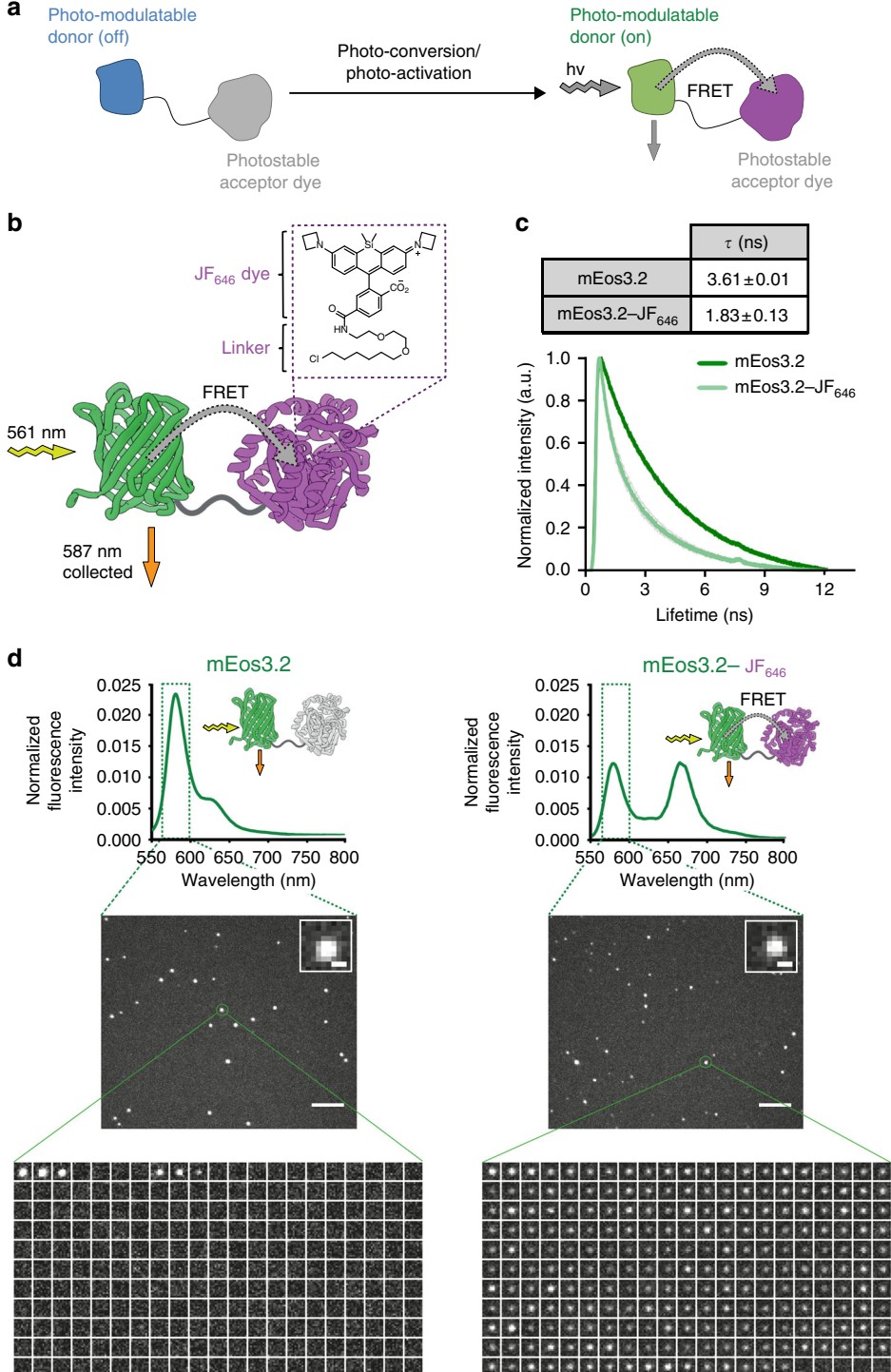

**Fig. 1** A single-molecule FRET pair containing mEos3.2 and JF$_{646}$. **a** The concept of using a photo-modulated fluorophore fused to a photostable acceptor. **b** A fusion protein was used as a test system to tether the JF$_{646}$ dye (via a HaloTag protein) to mEos3.2. **c** Fluorescence lifetime decay curves (background corrected and normalized to the maximum photon counts of each trace). The measured fluorescence lifetimes (mean ± standard deviation, see inset) decreased for mEos3.2 in the presence of the JF$_{646}$ dye. **d** Bulk emission spectra of mEos3.2 in the absence and presence of JF$_{646}$. Representative single-molecule traces of mEos3.2 at the 587 nm donor emission, depicted here as a montage of frames over time from left to right and top to bottom, showed reduced intensity and a longer on-time in the presence of the JF$_{646}$ dye (scale bar = 5 μm)

state switching events for each molecule (Fig. 2a, Supplementary Table 3). We also calculated the $k_{on}/k_{off}$ ratio[28], a metric of interest to those carrying out super-resolution experiments (Supplementary Table 3). Only the emission of mEos3.2 was characterized, because donor excitation also led to direct excitation of the JF$_{646}$—resulting in a high pre-conversion background

signal in the JF$_{646}$ acceptor emission channel (Supplementary Fig. 2c) that would prevent imaging at high density.

The quantification of these photophysical parameters allowed a comparison of mEos3.2 with and without JF$_{646}$. We found that the mEos3.2 molecules showed a mean increase in total on-state time of 1.5-fold compared to the unlabeled mEos3.2 molecules

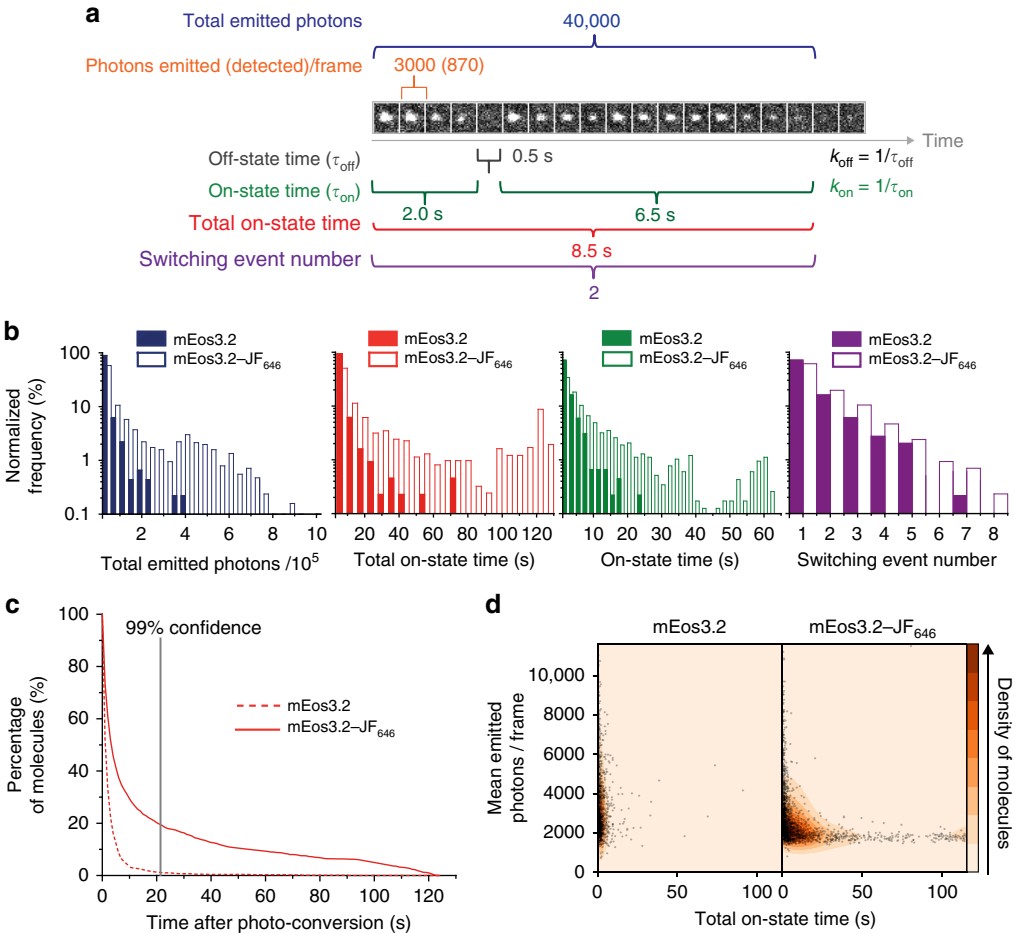

**Fig. 2** Improved photophysics of FRET-enhanced mEos3.2. **a** A representative trace of mEos3.2–JF$_{646}$ is shown as a montage of single frames over time. Photons detected were converted to photons emitted per on frame and summed to determine the total photon emission. The on-state time was calculated per switching event and summed to generate the total on-state time. The number of switching events was also calculated. **b** Histograms of total emitted photons, total on-state time, on-state time (i.e., individual track length), and total number of switching events per molecule, when performing single-molecule TIRF imaging using either mEos3.2 (filled bars) or the mEos3.2–JF$_{646}$ FRET pair (open bars) under identical imaging conditions. **c** The increase in total on-state time that occurred upon addition of the JF$_{646}$ dye is shown with a vertical line to illustrate that more than 20% of the mEos3.2–JF$_{646}$ molecules are still on when 99% of the mEos3.2 molecules are off. **d** At a single-molecule level, molecules with the longest total on-state times also had the lowest average intensity (i.e., highest FRET efficiency) measured as emitted photons/frame

(Supplementary Table 3). Given the reduction in fluorescence lifetime, we would theoretically expect a 2-fold increase in total on-state time[29] for mEos3.2. However, the total number of emitted photons also increased 1.7-fold suggesting that other factors also play a role in the photophysics of the mEos3.2–JF$_{646}$ FRET pair.

We hypothesized that transfer of energy to the acceptor JF$_{646}$ dye reduced the photobleaching rate of the donor fluorophore[18]. In an attempt to further control this, we also attempted to reduce the photobleaching rate of the acceptor JF$_{646}$ dye arising from either direct or FRET-based excitation. We therefore investigated whether Trolox, a triplet-state quencher, known to reduce the photobleaching of organic dyes in single-molecule and super-resolution imaging[30,31], would indirectly reduce photobleaching of the mEos3.2 donor fluorophore. Trolox was chosen because of its suitability for live-cell imaging: it is membrane permeable, non-cytotoxic, and may even improve cell viability[32]. We found that Trolox significantly increased the total number of photons emitted and the total on-state time before photobleaching of mEos3.2–JF$_{646}$ (Supplementary Table 3, Supplementary Movies 1 and 2). We determined the number of reversible on- to off-state switching events and showed that, under typical imaging

conditions, there was little change in the mean switching event number of 1.56 for mEos3.2 to 1.74 for mEos3.2–JF$_{646}$ (Fig. 2b, Supplementary Table 3). This switching event number was similar to values reported in previous studies[33,34]. The improved photophysical properties of mEos3.2 resulted from an increase of around 7.2-fold in the on-state time, which represents the useful time an individual molecule can be tracked experimentally and thus the likelihood of detecting low-probability events such as changes in the motion of a molecule (Fig. 2b, Supplementary Table 3). The >7-fold increase naturally also occurred for the total on-state time before photobleaching, which represents the total time a molecule can be imaged. This increase in total on-state time led to a 4.7-fold increase in the total photon budget from 20,000 ± 1000 to 94,000 ± 4000 photons emitted in the presence of JF$_{646}$ (Fig. 2b, Supplementary Table 3). The number of emitted photons for mEos3.2 matched well with a previously measured value of 21,000 photons for EosFP[35], and the value for mEos3.2–JF$_{646}$ is comparable to that previously measured for GFP (100,000 photons per molecule)[36]. This increase in photon budget was observed despite the reduction in mEos3.2 intensity (emitted photons in a single frame) of 39 ± 2 %.

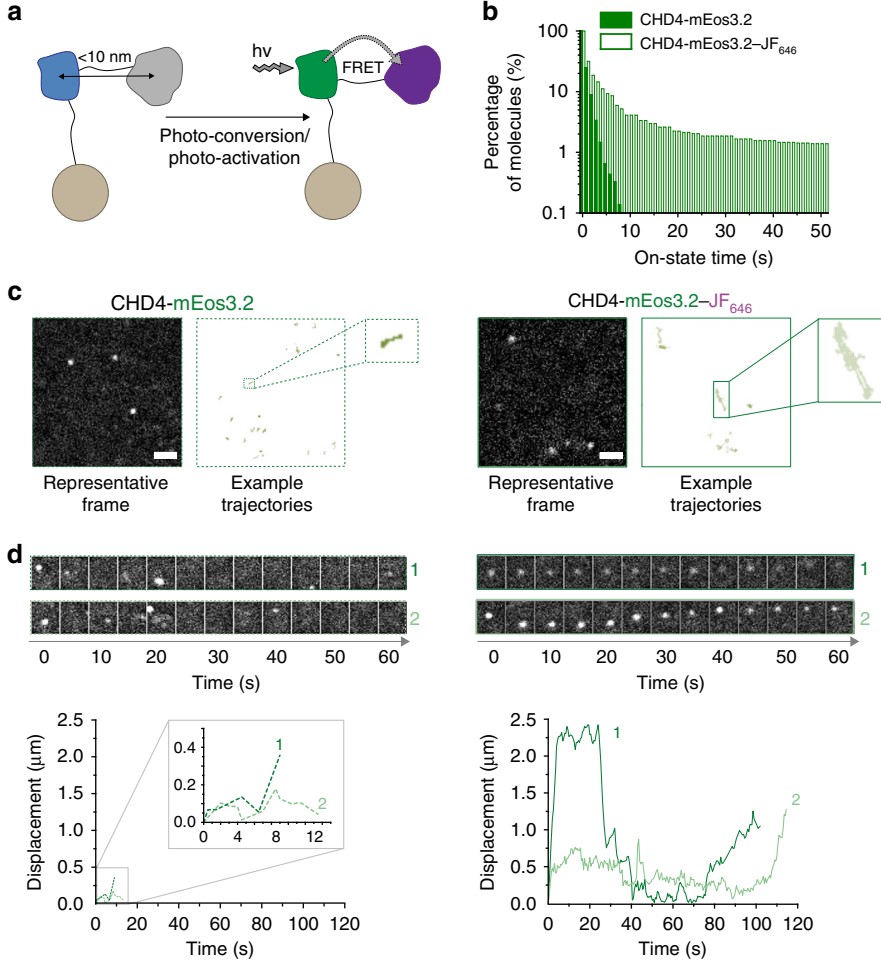

**Fig. 3** FRET-enhanced mEos3.2 improves tracking of single CHD4 proteins. **a** Schematic illustrating an experiment where the chromatin remodeler CHD4, a component of the NuRD complex, was fused with a C-terminal mEos3.2-HaloTag in mouse ES cells. **b** Histograms showing the percentage of molecules with a particular on-state time (i.e., individual track length) remaining after photo-conversion when performing sptPALM of either mEos3.2 (filled bars) or the mEos3.2–JF$_{646}$ FRET pair (open bars) under identical imaging conditions. **c** Representative 500 ms exposure images from the middle of the nucleus (left) and representative trajectories (right) are shown for mEos3.2- and mEos3.2–JF$_{646}$-tagged single CHD4 molecules (scale bar = 1 μm). **d** Montages (generated for every 10 frames collected) and plots of the distance moved from their point of photo-conversion, for two mEos3.2- and two mEos3.2–JF$_{646}$-tagged CHD4 molecules

At the single-molecule level, it was clear that many mEos3.2–JF$_{646}$ molecules emitted fluorescence for considerably longer times than others. After 99% of mEos3.2 molecules were photobleached, more than 20% of mEos3.2–JF$_{646}$ molecules were still in the on-state (Fig. 2c). There was considerable variety in the mean number of photons detected in each frame and those molecules that showed the highest FRET (lowest donor emission intensity) also showed the longest total on-state times (Fig. 2d). This suggested that if the FRET level were consistently higher, it could have resulted in even greater enhancements of the mEos3.2 on-state time. However, a balance needs to be struck because the low intensity peaks become increasingly difficult to distinguish from noise (Supplementary Fig. 3).

**FRET enhancement allows the recording of longer trajectories.** To demonstrate that single-molecule FRET can be used to track proteins in live mammalian cells, we designed two types of experiments. First, we tagged a single protein with both the donor PM fluorophore and the acceptor JF$_{646}$ dye such that every PM fluorophore detected can undergo FRET (Fig. 3a). Secondly, to study protein–protein interactions and track protein complexes,

we tagged one protein molecule with the donor PM fluorophore and another with the acceptor JF$_{646}$ dye (Fig. 4a). In both types of experiments, we showed that the photophysical characteristics improved as observed previously in vitro.

For the first experiment, knock-in cell lines were generated in which we tagged the chromatin remodeler CHD4 at the C-terminus with the mEos3.2-HaloTag fusion protein in mouse embryonic stem (ES) cells[37] (Fig. 3b). CHD4 was chosen because we had previously implemented cell viability assays to demonstrate that its function was unaffected by the presence of a tag (CHD4 null cells are not viable). CHD4 regulates ES cell pluripotency as part of the larger nucleosome remodeling and deacetylase (NuRD) complex. It also has roles in the DNA damage response and in cell cycle regulation[38]. It is widely distributed at high density throughout the nucleus and we imaged single mEos3.2-HaloTag-tagged CHD4 molecules, both with and without JF$_{646}$, using oblique-angle illumination and tracked their motion in 2D[6,37,39]. As rapidly diffusing CHD4 molecules can move out of the narrow depth of focus during 2D imaging, we focused on detecting the slower moving, presumably chromatin-bound, CHD4 molecules using a principle whereby freely diffusing molecules are blurred during a 500 ms

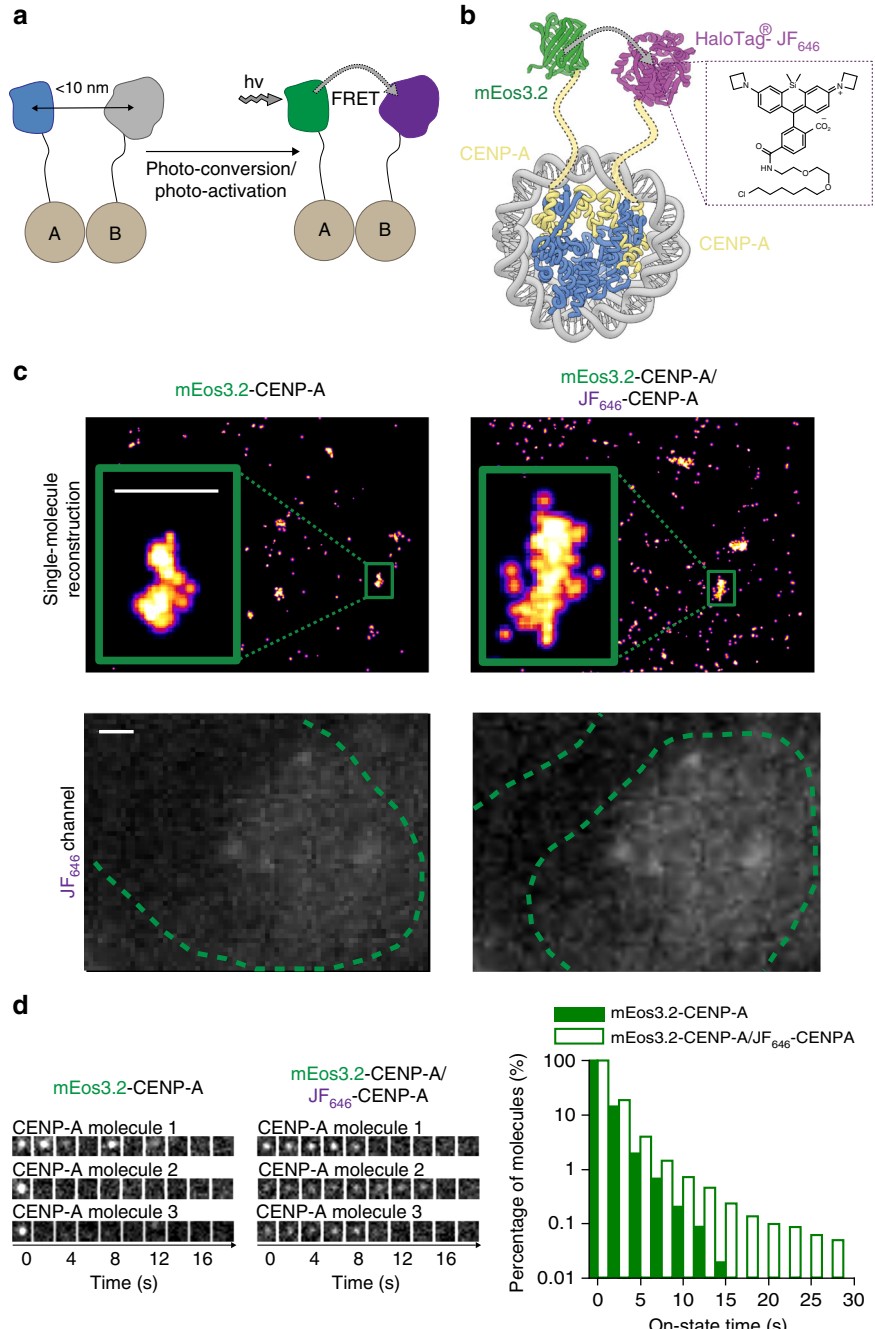

**Fig. 4** mEos3.2–JF$_{646}$ allows tracking of single CENP-A protein complexes. **a** Schematic of labeling one protein with a donor PM fluorophore and a second protein with JF$_{646}$ such that FRET occurs when they are in close spatial proximity. **b** Tagging CENP-A with either mEos3.2 or the HaloTag protein in the same cell allows assembly of nucleosomes where FRET can be observed between mEos3.2 and JF$_{646}$ in the (CENP-A/Histone H4)$_2$ hetero-tetrameric core. **c** (top) Representative reconstructions of single mEos3.2-tagged CENP-A nucleosomes localized at centromeres (see inset) are shown in the absence or presence of JF$_{646}$-labeled CENP-A. Scale bar = 500 nm. (lower) Images of the JF$_{646}$ dye show that the HaloTag-tagged CENP-A has been successfully labeled. Scale bar = 1 μm. **d** Single mEos3.2-tagged CENP-A nucleosomes show decreased intensity and increased track length in the presence of JF$_{646}$-tagged CENP-A. (left) Montages of three representative 500 ms exposure single-molecule traces (one image for every four frames) are shown in the absence or presence of the JF$_{646}$-labeled CENP-A. (right) Histograms showing the percentage of molecules remaining with a particular on-state time (i.e., individual track length) after photo-conversion when performing sptPALM in mouse ES cells. Cells expressing mEos3.2-tagged CENP-A are compared with cells also expressing JF$_{646}$-tagged CENP-A

exposure, but the more immobile chromatin-bound molecules remain localized[40,41]. We demonstrated that it was indeed possible to detect and localize mEos3.2–JF$_{646}$-tagged CHD4 molecules despite the reduction in mEos3.2 intensity from FRET in mEos3.2–JF$_{646}$. The precision at which mEos3.2 was localized was reduced from 15 to 28 nm in the presence of the JF$_{646}$, but

was still adequate for single-molecule localization and tracking (Supplementary Fig. 4).

In accordance with our in vitro experiments, comparison of mEos3.2- and mEos3.2–JF$_{646}$-tagged CHD4 molecules also showed a considerable increase in trajectory length (Fig. 3b). Both the short and long trajectories of bound CHD4 were mostly

immobile as expected for chromatin-bound molecules (Fig. 3c and Supplementary Movie 3). However, the longer trajectories allowed us to observe that the confined diffusion was interrupted by periods of rapid linear motion (Fig. 3d and Supplementary Movie 3). This could be due to movement as a result of ATP-dependent chromatin remodeling[38] or, as reported in previous studies[42,43], large-scale movements of the chromatin itself. Further work is necessary to understand what underlies the large movements of some of the CHD4 molecules. These experiments demonstrated that the mEos3.2–JF$_{646}$ FRET pair allows tracking of single chromatin-bound CHD4 molecules throughout the nucleus for extended periods of time—thereby increasing the likelihood of detecting low-probability changes in their movement.

We next tested whether our PM single-molecule FRET pair can also be used to track protein complexes for longer. We chose to study CENP-A where two CENP-A/Histone H4 dimers assemble to form a tetramer within the nucleosome core[44]. In this complex, the N-termini of the two CENP-A molecules are close in space (Fig. 4b) and we have shown in previous work that N-terminal tagging with FPs does not interfere with CENP-A function[34]. We expressed mEos3.2- and HaloTag-tagged CENP-A in mouse ES cells (Fig. 4c), which led to the formation of CENP-A nucleosomes containing either two mEos3.2-tagged molecules, one mEos3.2- and one JF$_{646}$-tagged molecule, or two JF$_{646}$-tagged molecules. The proportion of nucleosomes containing both mEos3.2- and JF$_{646}$–tagged CENP-A was also reduced by the presence of the untagged endogenous protein. To increase the likelihood of mEos3.2-tagged CENP-A molecules exhibiting FRET to JF$_{646}$-tagged molecules, a 5-fold excess of HaloTag-tagged CENP-A was expressed. When we carried out single-molecule imaging of mEos3.2-tagged CENP-A in the presence and absence of JF$_{646}$–tagged CENP-A (Fig. 4c), we found that single mEos3.2-tagged CENP-A molecules had decreased intensity, as would be expected for FRET, and longer track lengths in the presence of JF$_{646}$-tagged CENP-A (Fig. 4d, Supplementary Movies 4 and 5). This provided clear evidence of CENP-A/CENP-A interactions, but to rule out the possibility of artifacts due to non-specific binding of the JF$_{646}$ ligand, we also imaged cells where the JF$_{646}$ ligand was added in the presence of free HaloTag enzyme (i.e., where it was not fused to CENP-A). In those cells, we did not observe increased track lengths for mEos3.2-tagged CENP-A (Supplementary Fig. 5a, b). As expected, only a low percentage of mEos3.2-tagged CENP-A molecules exhibited extended track lengths, but the success of this experiment demonstrated that it is possible to use mEos3.2–JF$_{646}$ FRET to track single protein complex molecules.

To further show that FRET between mEos3.2 and the JF$_{646}$ dye can be used to track single protein complex molecules, we also tested whether FRET could be observed between mEos3.2-tagged CENP-A and the JF$_{646}$ dye attached to histone H2B via the SNAP tag (Supplementary Fig. 5c). In the nucleosome structure, the N-termini of CENP-A and histone H2B molecules are also close in space and N-terminal tagging of histone H2B with a SNAP tag has previously been reported to not interfere with its function[45]. We expressed both mEos3.2-tagged CENP-A and SNAP-tagged histone H2B in mouse ES cells (Supplementary Fig. 5d), which led to the formation of CENP-A nucleosomes with either one or two mEos3.2-tagged molecules alongside one or two JF$_{646}$-tagged histone H2B molecules. In this complex, FRET could occur between mEos3.2-tagged CENP-A and either the proximal or distal histone H2B molecules. We observed an increase in the track length of mEos3.2-tagged CENP-A in the presence of JF$_{646}$-tagged histone H2B (Supplementary Fig. 5e), but the increase was not as pronounced as that observed between CENP-A molecules. This is consistent with the fact that the average distance between

the N-termini of the CENP-A and histone H2B molecules (proximal and distal) is greater than that observed between the two CENP-A molecules within a nucleosome[44].

**Second FRET pair PA-JF$_{549}$–JF$_{646}$ further improves tracking.** To illustrate the generality of our approach, we investigated a second PM FRET pair by replacing the mEos3.2 photo-convertible donor fluorophore with the recently characterized photo-activatable dye PA-JF$_{549}$[17] (Supplementary Fig. 6a). A similar fusion protein was expressed and purified, but where the PA-JF$_{549}$ dye was attached to the SNAP-tag protein (in place of mEos3.2). PA-JF$_{549}$ is brighter than mEos3.2[17], with a higher quantum yield (Supplementary Fig. 1a) and it too has good spectral overlap with JF$_{646}$ (Supplementary Fig. 1e). We confirmed FRET by measuring the fluorescence lifetime for PA-JF$_{549}$. In the absence of JF$_{646}$, we found a mean excited-state lifetime of 2.9 ns, but upon addition of JF$_{646}$ this decreased to 2.1 ns (Supplementary Fig. 6b). This decrease in excited-state lifetime corresponded to a FRET efficiency of $0.28 \pm 0.05$ and a mean inter-fluorophore distance of $7 \pm 3$ nm (see Methods). Bulk fluorescence emission spectra were also collected to demonstrate a reciprocal decrease in the donor and increase in the acceptor fluorescence (Supplementary Fig. 6c). These results confirmed that PA-JF$_{549}$–JF$_{646}$ undergoes FRET after photo-activation. Further confirmation was achieved by carrying out photophysical analysis of PA-JF$_{549}$–JF$_{646}$ in our optimized Trolox conditions after immobilizing the relevant fusion proteins on a glass cover-slip at low spatial density (Supplementary Fig. 7a, b). We observed a $19 \pm 10\%$ reduction in PA-JF$_{549}$ intensity in the presence of the JF$_{646}$ dye, close to the value expected from our FRET efficiency calculation above. As with mEos3.2, we did not observe a significant change in the number of switching events (Supplementary Fig. 7c). We observed a 2.3-fold increase in the on-state time, as well as 2.0- and 1.7-fold increases, respectively, in the total on-state time and photon budgets of PA-JF$_{549}$ in the presence of JF$_{646}$ (Supplementary Fig. 7). Although there was a lower relative increase compared to mEos3.2, these data confirm the generality of our approach for enhancing the photophysical properties of PM fluorophores.

To show that the PA-JF$_{549}$ donor fluorophore can be used for improved live-cell tracking of protein complexes, we expressed SNAP-tagged histone H2B alone in mouse ES cells (Fig. 5a). Each nucleosome contains two histone H2B molecules, hence FRET can either occur within a nucleosome or between nucleosomes. However, because the histone H2B molecules are on opposing sides of the nucleosome, a greater FRET efficiency is observed due to inter-nucleosomal FRET as opposed to intra-nucleosomal FRET. As a result, a higher level of FRET was observed when chromatin was compacted either in regions of heterochromatin or in cells undergoing mitosis[46] (Fig. 5a). We carried out single-molecule localization and tracking of PA-JF$_{549}$-tagged H2B, both with and without JF$_{646}$-tagged H2B. By adding the PA-JF$_{549}$ SNAP tag ligand in the presence of a ten-fold excess of JF$_{646}$ SNAP tag ligand, the likelihood that PA-JF$_{549}$-tagged H2B molecules were in close proximity with a neighboring JF$_{646}$-tagged H2B was increased. As expected for molecules undergoing FRET, we observed a decrease in the integrated fluorescence signal of some of the PA-JF$_{549}$-tagged H2B molecules (Fig. 5b), which was also reflected by a modest decrease in the mean localization precision from 11.9 to 13.3 nm (Supplementary Fig. 4). In addition to a change in signal intensity, we also observed a significant increase in the trajectory length of PA-JF$_{549}$-H2B in the presence of JF$_{646}$ (Fig. 5c).

To identify PA-JF$_{549}$-tagged H2B molecules undergoing FRET, and thereby only track protein complexes, we filtered trajectories

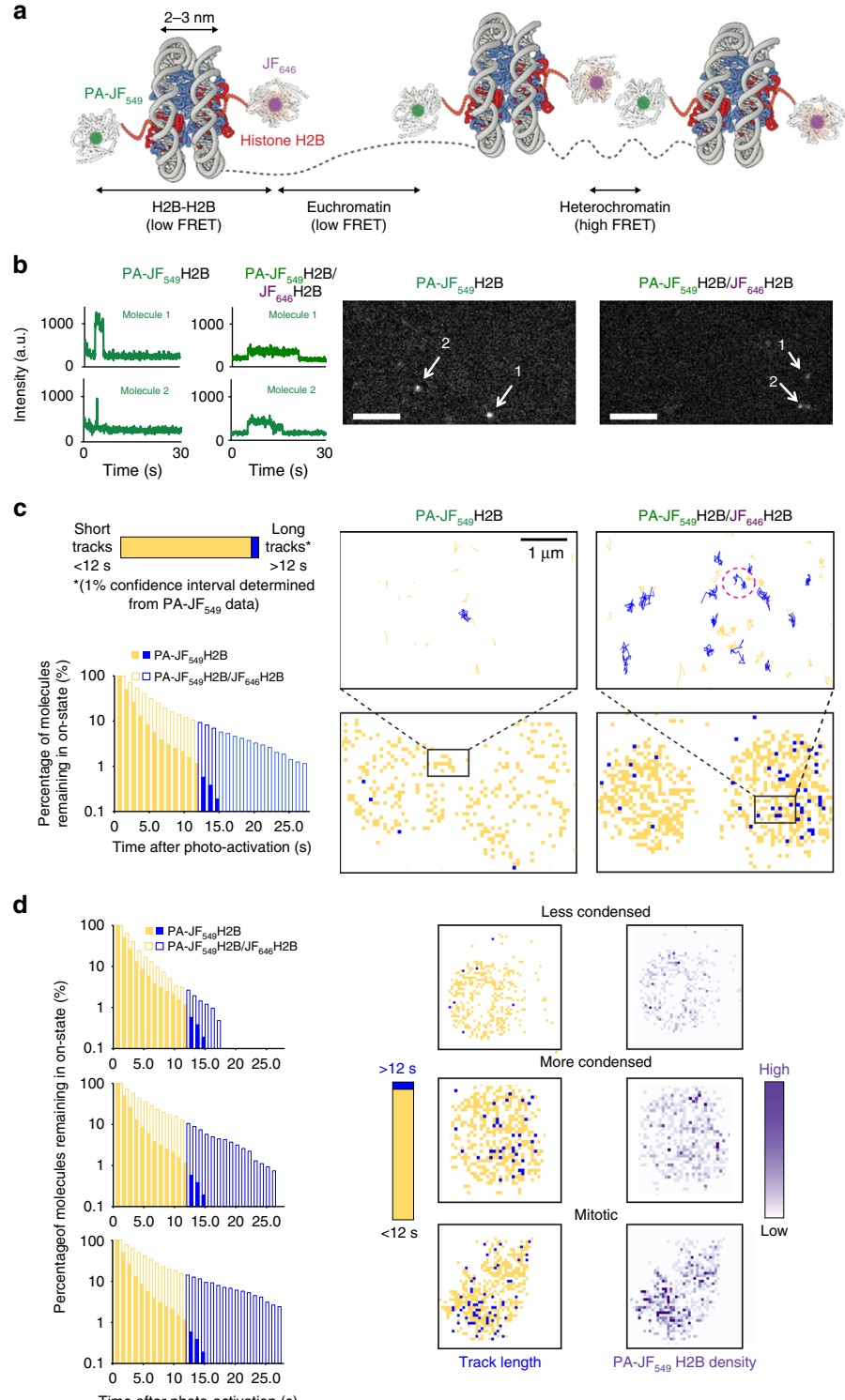

**Fig. 5** FRET-enhanced PA-JF$_{549}$ allows tracking of single H2B complexes. **a** Schematic showing one histone H2B molecule in a nucleosome labeled with a donor PA-JF$_{549}$, and a second labeled with an acceptor JF$_{646}$ such that FRET occurs either within a nucleosome or between nucleosomes when they are in close spatial proximity. The reduced inter-nucleosome distance in heterochromatin leads to higher FRET efficiency. **b** Representative images and intensity vs. time traces of single PA-JF$_{549}$-tagged H2B molecules are shown in the absence or presence of JF$_{646}$-labeled H2B (scale bar = 5 μm). **c** Single PA-JF$_{549}$-tagged H2B nucleosomes show increased track length in the presence of JF$_{646}$-tagged H2B. A 1% confidence interval was defined to show whether average trajectory lengths were below (yellow) or above 12 s (blue)—only 1% of PA-JF$_{549}$-tagged molecules have average track lengths longer than 12 s. Density maps were generated to show where the long trajectories are concentrated, and representative examples of long trajectories are shown in the upper panels in the same colors, where the one circled in red shows long periods of confined motion interrupted by a short period of rapid linear motion. **d** Inter-nuclear differences in track length were observed (left) for three different cells with less condensed, more condensed, or mitotic chromatin (top to bottom). Density maps show that regions where track lengths were on average above the 12-s cutoff (middle) also had a high number of localized PA-JF$_{549}$-tagged H2B molecules (right)

by their track length (Fig. 5c). There was a 10% likelihood of trajectories being longer than 12 s in the presence of JF$_{646}$, but only a 1% likelihood in the absence of JF$_{646}$. When we mapped the trajectories longer than 12 s onto individual cells, we found that the long tracks were mostly in cells with greater H2B density such as those undergoing mitosis (Fig. 5d). In cells with lower H2B density, long trajectories were localized to regions with a higher relative density of chromatin (heterochromatin). Studying these longer trajectories showed that some H2B molecules underwent periods of confined diffusion followed by sudden bursts of rapid motion. Our results are consistent with previous imaging of H2B clusters[47] but, as with the results for CHD4, more work is required to understand the biological implications of these movements.

Finally, from the observed reduction in intensity of PA-JF$_{549}$-tagged H2B molecules in the presence of JF$_{646}$-tagged H2B, we determined the FRET efficiency of individual molecules (Supplementary Fig. 8). It is difficult to study FRET efficiency in the presence of high levels of FRET because the molecules also have low intensity. This precluded an analysis of mEos3.2–JF$_{646}$ FRET efficiency because the dimmer mEos3.2 offers a relatively smaller dynamic range over which a reduction in intensity can be measured before the molecule is too dim to be localized for single-molecule imaging. However, when we carried out an analysis of FRET efficiency for PA-JF$_{549}$, using the average intensity of a trajectory relative to that expected in the absence of FRET (i.e., in cells without JF$_{646}$), we observed three dominating FRET states in most cells, suggesting three different levels of H2B compaction. It was also clear that H2B/H2B FRET efficiency increased in cells undergoing mitosis such that only the highest FRET state could be observed in these cells (Supplementary Fig. 8). Similar results have previously been reported using FLIM[46].

## Discussion

We describe here a novel approach for tracking single proteins and protein complexes in live cells for extended periods of time. By placing an organic acceptor dye in close spatial proximity to a PM fluorophore, we have developed novel single-molecule FRET pairs that exploit the well-studied and useful characteristics of mEos3.2 and PA-JF$_{549}$. The design of these FRET pairs has also allowed us to utilize existing tools (such as a triplet-state quencher) to enhance the photostability of the solvent-accessible JF$_{646}$ acceptor dye, and thus impart changes to the photostability of the donor PM fluorophores.

Although much is known about externally modifying the photophysical properties of organic dye molecules for single-molecule imaging[30,31,48,49], there are currently few ways of altering those of PM FPs, almost certainly due to the relative inaccessibility of the chromophore in the interior of the β-barrel of the protein[50,51]. Since most PM fluorophores are currently still fluorescent proteins, our approach offers a way to positively modify them while preserving their specific properties of photo-activation/conversion. Here, we have demonstrated a novel approach for increasing the track length and photon budget of PM FPs. These improvements result from the selective control of excited-state kinetics; when an electron in the PM FP is excited, multiple kinetic pathways become active—i.e., fluorescence, conversion into long/short dark states, and photobleaching[50]. However, in the presence of FRET, energy transfer to the acceptor fluorophore is favored. At this point, there are again multiple pathways for return to the ground state. Previous studies have shown that photobleaching of mEos3.2 mainly occurs through a mechanism that is likely to involve the triplet state[52]. We therefore anticipate that mEos3.2 photobleaching (in the

mEos3.2–JF$_{646}$ FRET pair) is reduced by energy transfer from the mEos3.2 excited state to JF$_{646}$ via FRET, which thereby reduces the probability of formation of the mEos3.2 triplet state. Similar mechanisms have previously been suggested[53]. The exposure of the acceptor to the solvent also means that the FRET system can now be manipulated using approaches that modify the properties of the acceptor. Here, we used Trolox to quench the triplet state of JF$_{646}$[30], which in turn keeps the FRET pathway active and thereby increases the photostability of the PM donor. For mEos3.2, this resulted in a 7-fold increase in on-state time, and a 4.7-fold increase in photon budget. Our approach of using a solvent-accessible FRET acceptor dye to positively modify the photophysical properties of the chromophore within FPs should be useful for modifying the properties of other PM FPs.

To demonstrate that our FRET approach has broad applicability, we extended it to improve the photophysical properties of the recently characterized photo-activatable dye PA-JF$_{549}$[17]. Although the photobleaching mechanism of PA-JF$_{549}$ is less well understood, we expect that similar mechanisms may operate. There was less of a comparative improvement in the photon budget of PA-JF$_{549}$, but because PA-JF$_{549}$ can emit more photons in a given time period than mEos3.2, it is likely to be useful when studying the dynamics of molecules that must be localized at high precision or to determine FRET efficiencies.

Recent work has illustrated the power of sptPALM to study the interactions of nuclear proteins and protein complexes with chromatin[4,7]. In the presence of Trolox, we have shown that our mEos3.2–JF$_{646}$ and PA-JF$_{549}$–JF$_{646}$ FRET pairs can be used to improve such studies by increasing the mEos3.2 and PA-JF$_{549}$ trajectory lengths on both single proteins and protein complexes in live mammalian cells. When tracking protein complexes a tag needs to be added to each protein (as in previous FRET approaches), but when tracking single proteins our approach uses a larger tag than either mEos3.2 or HaloTag alone, which may not always be suitable. Of course, our approach also leads to a decrease in intensity of the PM fluorophore and it is therefore not appropriate in situations where localization precision is key. In these situations, using either the mEos3.2 or PA-JF$_{549}$ dye alone would be preferable. As in conventional sptPALM, where lower excitation powers are often used to increase the trajectory length at the expense of localization precision, our method trades-off localization precision for a more extended trajectory length. For example, when using the motion blurring experiments to either estimate the time a protein binds to chromatin or to study its motion on chromatin, a high localization precision is less important as long as there is sufficient signal-to-noise for detection and tracking. Notably, however, in contrast to conventional sptPALM where a doubling of trajectory length is achieved by halving the power density, and hence intensity, of the molecule, with FRET we observe a 7-fold increase in mEos3.2 on-state time with a halving of its intensity as a result of altering the standard photobleaching pathways of the PM donor.

Our approach now allows us to record longer trajectories to detect rare transitions between constrained and rapid linear motion and we observed such transitions for both bound CHD4 and histone H2B molecules in nucleosomes. Previous evidence for directed movement of histone H2B involved reconstructing the average positions of H2B clusters using a sliding window that combined the localizations of several H2B molecules within a cluster[47], but here we are able to directly study single molecules. In our current experiments, we have focused on studying relatively stable chromatin binding, but future studies using a 3D microscope will be able to take advantage of the longer track lengths for studying transitions between bound and freely diffusing molecules.

We anticipate that our FRET pairs will prove particularly useful for understanding the dynamics of single protein complex molecules. Here, we studied the formation of specific protein complexes in the nucleus and the cell cycle stages at which they form. However, the approach may now be used to study how dynamics or residence times of specific protein complexes differ from those of the isolated protein or other complexes containing the same protein. Although multi-color co-localization approaches have been developed[3,54], they are difficult to implement where there is either a high density of the individual proteins or other complexes containing the same proteins, as occurs with CENP-A at centromeres in eukaryotic cell nuclei. Complementation methods (utilizing a split version of mEos3.2[10]) are also adversely affected by densely packed environments because the split proteins can themselves facilitate interactions between the tagged proteins. More importantly, complementation approaches do not provide a means to increase the track lengths of protein complexes, one of the major appeals of our approach. Finally, we envisage that imaging the acceptor fluorophore of FRET pairs may prove useful as a biosensor in the detection of protein–protein interactions using stochastic optical fluctuation imaging[9].

## Methods

**Protein structure analysis**. Data from published crystal structures were analyzed using the UCSF Chimera package[55]. The Chimera software was also used to generate figures for this paper. A list of the protein structures analyzed are as follows: mEos3.2 (PDB ID: 3P8U)[56], HaloTag (PDB ID: 4KAA)[57], SNAP tag (PDB ID: 3KZZ)[58], and the CENP-A nucleosome (PDB ID: 3AN2)[44].

**Expression of mEos3.2-HaloTag and HaloTag-SNAP-tag proteins**. A mEos3.2-HaloTag construct was cloned from plasmids containing mEos3.2[59] and HaloTag (Promega), fused via a short linker (Leu-Glu-Gly-Ser), and inserted into an EcoRI/HindIII digested pET30a expression vector using an In-Fusion HD cloning kit (Clontech). A HaloTag-SNAP construct was cloned in the same way. Site-directed mutagenesis of the mEos3.2-HaloTag vector was then carried out to generate the fusion construct with a different linker length (ΔARRELEGSE).

His-tagged mEos3.2-HaloTag and HaloTag-SNAP constructs were expressed in *E. coli* BL21(DE3)pLysS cells by growing a liter of cells from a 50-ml starter culture to an $OD_{600\,nm}$ of 0.7–1.0 in LB media containing 35 μg/ml kanamycin and 34 μg/ml chloramphenicol before inducing expression using 1 mM IPTG (Melford Laboratories Ltd) for 4 h at 25 °C. Cells were pelleted and stored at −20 °C. For protein purification, cells were thawed rapidly at 37 °C and resuspended in three volumes of lysis buffer (50 mM Hepes pH 7.5, 300 mM NaCl, 5% glycerol and protease inhibitor cocktail (Roche)) per volume of pellet. The cell resuspension was then sonicated for 13 min at 33% amplitude (5 s on and 10 s off) using a Sonic Dismembrator (Model 505, Fisher Scientific), the cell debris was pelleted and the protein lysate collected and filtered through a 0.45-μm filter.

The fusion proteins were first purified by nickel affinity chromatography. The protein lysate was resuspended in 5 ml of 50% Ni-nitrilotriacetic acid (Ni-NTA) bead slurry (Qiagen), which had been pre-equilibrated in buffer (50 mM $NaH_2PO_4$, 300 mM NaCl, pH 8.0), and was rotated overnight at 4 °C. The fusion protein was purified using a 20-ml gravity column (Bio-Rad), washed in three column volumes of wash buffer (50 mM $NaH_2PO_4$, 300 mM NaCl, 5 mM imidazole, pH 8.0) and finally eluted in 10 ml of elution buffer (50 mM $NaH_2PO_4$, 1 M NaCl, 250 mM imidazole, pH 8.0). Protein was then dialyzed (into 50 mM $NaH_2PO_4$, 1 M NaCl, pH 8.0) to remove imidazole using a Vivaspin 500 with a molecular weight cutoff (MWCO) of 5 kDa (Sartorius Stedim Biotech). The fusion protein was then further purified by gel filtration in a buffer consisting of 50 mM $NaH_2PO_4$ and 1 M NaCl (75 mL Sephadex200 column, GE Healthcare). Mass spectrometry was carried out to verify the mass of the protein (65096.8 Da) and amino acid analysis was used to determine its concentration. The protein was stored in 500 μl aliquots of 1 mg/ml at −80 °C and re-purified using size exclusion chromatography prior to dye labeling. The fusion proteins (5–10 μM) were labeled with the HaloTag-dye or SNAP tag-dye ligands by reacting them at an equi-molar ratio at room temperature for 1 h before purifying using an Illustra Nap-5 gravity column (GE Healthcare). Labeling was confirmed to be close to 100% by mass spectrometry.

The HaloTag or SNAP-tag PA-JF_549 and JF_646 dyes were a kind gift from Luke D. Lavis (HHMI)[60].

**Bulk fluorescence spectra characterization**. Bulk fluorescence spectra were obtained with mEos3.2 or PA-JF_549 with or without the JF_646 dye (15–25 μM) using a fluorescence spectrophotometer (Cary Eclipse). The sample was placed in a quartz cuvette (Hellma Analytics, 3×3 mm). To determine the actual FRET

efficiency of the mEos3.2-HaloTag proteins with the different dyes, emission spectra were collected after exciting at 532 nm and detecting fluorescence over a range of wavelengths (550–800 nm). FRET efficiency ($E$) was calculated using the standard equation (Eq. 1):

$$E = 1 - \frac{I_D{}'}{I_D},$$

where $I_D$ is the fluorescence intensity of the donor fluorophore alone (mEos3.2) and $I_D{}'$ is the fluorescence intensity of the donor in the presence of the acceptor (mEos3.2–JF_646).

The distance $r$ between the donor mEos3.2 or PA-JF_549 and acceptor JF_646 fluorophores was calculated using the equation for FRET efficiency ($E$) given below[61] (Eq. 2):

$$E = \frac{1}{1 + (r/R_0)^6}.$$

$R_0$ was calculated to be 6.01 nm for mEos3.2–JF_646 and 5.81 nm for PAJF_549–JF_646 from the following equation (Eq. 3):

$$R_0^6 = \frac{9\ln 10}{128\pi^5 N_A} \frac{\kappa^2 Q_D}{n^4} J,$$

where $N_A$ is Avogadro's number, $\kappa^2$ is the dipole orientation factor ($\frac{2}{3}$ for freely rotating donor and acceptor fluorophores), $Q_D$ is the quantum yield of the donor mEos3.2 or PA-JF_549 fluorophores (0.55 and 0.88, respectively), $n$ is the refractive index of the medium (1.33), and $J$ is the spectral overlap integral between the donors mEos3.2 or PA-JF_549 and acceptor JF_646 spectra calculated over different wavelengths $\lambda$ using the extinction coefficient $\epsilon_A$ of the acceptor JF_646 dye (max value of 152,000 $M^{-1}\,cm^{-1}$) (Eq. 4):

$$J = \int \bar{f}_D(\lambda)\epsilon_A(\lambda)\lambda^4 d\lambda.$$

**Fluorescence lifetime imaging**. Time-Correlated Single Photon Counting (TCSPC) measurements were performed on a Leica SP8 STED 3× system additionally equipped with Single Molecule Detection (SMD) software (SymPhoTime version 5.3.2.2) and hardware (PicoHarp 300; PHR 800) from PicoQuant. Fluorescence excitation in this system was with a pulsed (80 MHz) tuneable white light laser (WLL; Super-K; NKT Photonics) while fluorescence detection was with an internal hybrid single-molecule detector (Leica HyD SMD). Donor excitation was performed at a wavelength of 561 nm with a detection band of 570-620 nm and using a 20 × 0.75 NA water-immersion objective (HC PL APO CS2) with a zoom factor of 1× and a scan speed of 400 Hz, and with a frame accumulation of 25 images. Photo-activation/conversion was achieved using the 405 nm laser for 1 min at activation powers of 1.75 kW/cm² at the focal plane of the objective prior to the lifetime measurement. An increase in donor emission occurred post-activation/conversion as expected. FLIM measurements were performed in triplicate pre- and post-activation/conversion on purified protein in 50 mM $NaH_2PO_4$, 1 M NaCl, pH 8.0 in 8-well glass bottom μ-Slides (iBidi). Protein was attached to the coverslip by incubating with poly-L-lysine (Sigma Aldrich) for 30 min and then with 5–10 μM protein for 10 min. Buffer was replaced to remove floating protein and then imaging of the attached protein carried out by focusing on the coverslip. Attachment was validated by imaging to ensure that photo-converted molecules did not diffuse away over time.

Post-activation/conversion TCSPC decay data were analyzed using SymPhoTime software (version 2.1) from Picoquant. In this analysis, we analyzed the tail of the cumulative TCSPC decay data from all pixels in the image with a minimum number of 50 photons and with a raw time range of 1.5 ns ≤ τ ≤ 7.0 ns. The decay curve was fit to a sum of exponential tail decay curves defined by (Eq. 5):

$$I(t) = I_{Bkgd} + \sum_i \alpha_i e^{-t/\tau_i}$$

where $I_{Bkgd}$ is an intensity offset for the background counts, and $\alpha_i$ and $\tau_i$ are the amplitude and lifetime values for the $i$th exponential where we compared the fits for $1 \le i \le 3$ (i.e., mono-, bi-, and tri-exponential fits). We subsequently determined the relative likelihood for each model for each dataset by use of the Bayesian Information Criterion (BIC)[62,63]. In short, using this approach, we calculated the BIC for each model from (Eq. 6)

$$BIC = \ln[n](p + 1) + n\left(\ln\left[\frac{2\pi RSS}{n}\right] + 1\right),$$

where $n$ is the number of data points, $p$ is the number of free parameters for the fit, and RSS is the residual sum of squares of the fit. We subsequently determined the relative likelihood for each model for each dataset from (Equation 7):

$$\text{Relative Likelihood} = \text{Exp}\left[\frac{BIC_{Min}(\text{model}) - BIC(\text{model})}{2}\right].$$

The data for both mEos3.2 with and without JF$_{646}$ and PA-JF$_{549}$ with and without JF$_{646}$ can be described by two lifetimes. We determined the FRET efficiency ($E$) using the average fluorescence lifetimes (Eq. 8):

$$E = 1 - \frac{\tau_{DA}}{\tau_D},$$

where $\tau_D$ is the amplitude weighted average fluorescence lifetime of the donor fluorophore alone (mEos3.2 or PA-JF$_{549}$) and where $\tau_{DA}$ is the amplitude weighted average fluorescence lifetime of the donor in the presence of the acceptor (mEos3.2–JF$_{646}$ or PA-JF$_{549}$–JF$_{646}$).

**Microscope setup**. Two bespoke microscope setups were used in this work both of which are described below. A table detailing specific parameters used for each experiment can be found in the supplementary information.

Microscope 1: An IX71 Olympus inverted microscope was used with circularly polarized laser beams aligned and focused at the back aperture of an Olympus 1.49 NA 60× oil objective (Plan Apochromat 60× NA 1.49, Olympus APON 60XOTIRF). Continuous wavelength diode laser light sources used include a 561 nm (Cobolt, Jive 200, 200 mW) and a 405-nm laser (Oxxius, LaserBoxx 405, 100 mW). Total internal reflection was achieved by aligning the laser off axis such that the emergent beam at the sample interface was near-collimated and incident at an angle greater than the critical angle $\theta_c \sim 67°$ for a glass/water interface for TIRF imaging and slightly less than $\theta_c$ for oblique-angle illumination imaging. This generated a ~50-μm diameter excitation footprint. For TIRF, the power density at the coverslip for the 561 nm laser was calculated to be approximately 0.4 kW/cm$^2$ measured with the laser beam in epi-illumination. For oblique-angle illumination, the power of the collimated beams at the back aperture of the microscope was 10 kW/cm$^2$ and 10–100 W/cm$^2$ for the 561 nm and 405 nm laser beams, respectively. The lasers were reflected by dichroic mirrors, which also separated collected fluorescence emission from the TIR beam (Semrock, Di01- R405/488/561/635). The fluorescence emission was collected through the same objective and then further filtered using a combination of long-pass and band-pass filters (BLP01-561R and FF01-587/35 for 561 nm excitation). The emission signal was expanded through a 2.5× achromatic beam expander (Olympus, PE 2.5× 125) and finally projected onto an EMCCD (Photometrics, Evolve 512) with an electron multiplication gain of 250 ADU/photon operating in a frame transfer mode. The instrument was automated using the open-source software micro-manager (https://www.micro-manager.org) and the data displayed using the ImageJ software[64,65].

Microscope 2: An IX73 Olympus inverted microscope was used with circularly polarized laser beams aligned and focused at the back aperture of an Olympus 1.40 NA 100× oil objective (Universal Plan Super Apochromat, 100×, NA 1.40, UPLSAPO100XO/1.4). Continuous wavelength diode laser light sources used include a 641-nm (Coherent, CUBE 640–100 C, 100 mW), a 561-nm (Cobolt, Jive 200, 200 mW), and a 405-nm laser (Stradus, Toptica, 405–100, 100 mW). TIRF and oblique-angle illumination imaging were performed with identical dichroic mirrors and emission filters. The emission signal was projected onto an EMCCD (Photometrics, Evolve 512 Delta) with an electron multiplication gain of 250 ADU/photon operating in a frame transfer mode. The instrument was automated using the open-source software micro-manager (https://www.micro-manager.org) and the data displayed using the ImageJ software[64,65].

**TIRF characterization of mEos3.2-HaloTag-dyes**. Borosilicate glass coverslips (VWR Int, 22 × 22 mm) were cleaned to remove any fluorescent residues in an argon plasma cleaner (Harrick plasma) for 1 h. Frame-seal incubation chambers (Bio-rad) were attached to the coverslip and 50 μl of 0.1% poly-L-lysine (Sigma Aldrich) added to the center of the chamber for 30 min; 50 μl of 10 nM protein was then added to the poly-L-lysine-coated coverslip for 10–15 min. The sample was washed three times with 50 μl of filtered (0.2 μm syringe filter, Whatman, 6780–1302) MilliQ water and fluorescence images collected as movies of 500 images at 500 ms exposures. Photo-conversion was achieved as a single pulse in the first frame of each movie.

**Analysis of photophysical parameters**. The experiment was replicated in the laboratory twice on Microscope 1 and twice on Microscope 2. Although all experiments showed similar results, we chose to analyze datasets from Microscope 1 recorded on the same day to reduce systematic errors arising from subtly different microscope alignments. We analyzed data collected on the same day from Microscope 1 in which we tracked 455 and 454 single mEos3.2 and mEos3.2–JF$_{646}$ molecules without Trolox, 990 and 1568 single mEos3.2 and mEos3.2–JF$_{646}$ molecules with 2 mM Trolox. Given that 20% of single mEos3.2–JF$_{646}$ molecules were in the on-state considerably longer than single mEos3.2 molecules, this is an appropriate sample size to demonstrate the change we observe. All histograms were generated using the Origin package (OriginLab, Northampton, MA).

A brief description of the software used for data analysis is described here. A maximum-intensity projection of the first two images after photo-conversion was used as the basis for detecting single molecules. A Laplacian-of-gaussian filter was applied to the projection, and local maxima found. Scripts for this are available at https://github.com/TheLaueLab/blob-detection, and for all remaining steps at https://github.com/TheLaueLab/blink-analysis. A region centered on each peak

with a threshold value of >600 ADU was extracted from each image. This region consisted of a 7-by-7 pixel signal region, and a surrounding 2-pixel wide background region. Individual frames were simplified to 1D traces by subtracting the mean per-frame background from the corresponding frame, and then taking the mean of all pixels in that frame. A hidden Markov model (using the hmmlearn python package from https://github.com/hmmlearn/hmmlearn) was set up with four states: two on-states, one off-state, and one bleached state. Transitions were equally likely between all on and off states, and 1/10th as likely from any on state to the bleached state (transitioning away from the bleached state was impossible). The states were initialized with a mean of 300 ADU for the on-states and 0 for the off and bleached states, with a prior weight of 1e3 assigned to the state means. The model was trained on all traces from a particular fluorophore, and the same trained model was used to categorize all traces.

The total on-state time of each molecule was calculated by counting the number of images in an on-state. A blink was defined as a run of consecutive on images; the mean run length multiplied by the exposure time is the on-state time, and the number of runs detected for a particular molecule is the number of switching events. The off-rate was the number of blinks divided by the total on-state time, and the on-rate was the number of blinks divided by the total off-state time (excluding the last run of off-frames if it continued to the end of the video).

Finally, the total photon emission was calculated for each on image, subtracting the mean of the background region from the signal region. In order to calculate the number of emitted photons per molecule, the total camera gain in units of analog-to-digital units (ADU)/photon was determined by (Eq. 9)

$$G_{total} = \frac{1}{G_{camera}} \times G_{EM} \times QE,$$

where $G_{camera}$ is the signal amplification inherent in the EMCCD in units of ADU/electron, $G_{EM}$ is the ratio of the charge on the camera with and without gain, and QE is the quantum efficiency—the ability of the camera to produce a charge as a result of an incident photon with units of electrons/photon. $G_{total}$ is 33.1 ADU/photon and 35.7 ADU/photon for Microscopes 1 and 2, respectively.

The measured signal ($I$) in units of electrons was converted to emitted photons ($n$) as follows (Eq. 10):

$$n = \frac{I}{G_{total} \times TE}.$$

TE is defined as the transmission efficiency of all optical components in the emission path of the instrument and can be described by (Eq. 11)

$$TE = \eta_{coll} \times T \times \eta_{EMCCD},$$

where $\eta_{coll}$ is the collection efficiency of the objective, $T$ is the transmission of the internal optical components of the microscope, and $\eta_{EMCCD}$ is the quantum efficiency of the EMCCD[66].

**Mammalian cell culture and cell line generation**. ES cells were cultured in standard serum and mouse leukemia inhibitory factor (mLIF) conditions: Glasgow minimum essential medium (Sigma-Aldrich G5154) containing 100 mM 2-mercaptoethanol (Life tech, cat. 21985023), 1× Minimum Essential Media, non-essential amino acids (Sigma-Aldrich, M7145), 2 mM L-glutamine (Life tech, cat. 25030024), 1 mM sodium pyruvate (Sigma-Aldrich, S8636-100ML), 10% fetal bovine serum (HyClone FBS, Lot nr SZB20006, GE Healthcare Austria SV30180.03), and 10 ng/ml mLIF (provided by the Biochemistry Department, University of Cambridge). They were passaged every 2 days by washing in PBS (Sigma-Aldrich, D8537), adding Trypsin-EDTA 0.25% (Life tech, cat. 25200072) to detach the cells, and then washing in media before re-plating in fresh media. To help the cells attach to the surface, plates were incubated for 15 min at room temperature in PBS containing 0.1% gelatin (Sigma Aldrich, G1890). The background E14tg2a ES cell lines (available from Sigma Aldrich, 08021401) were characterized by qPCR, RNA-seq, ChIP-seq, and potency assays, and they were routinely screened for mycoplasma contamination and tested negative.

ES cells expressing mouse CHD4 tagged at the C-terminus with the mEos3.2-HaloTag were generated as by CRISPR/Cas9 based knock-in of a cassette containing mEos3.2-HaloTag and a puromycin selection gene into one CHD4 allele of the ES cells[37]. The puromycin cassette was then removed using Dre recombinase to generate the CHD4 allele with a C-terminal mEos3.2-HaloTag fusion. Since knockout of CHD4 is lethal, we used cell viability assays to verify that the function of the tagged CHD4 was unaffected. The E14tg2a ES cell line[67] expressing mEos3.2-tagged CENP-A has previously been described[59] but briefly, was generated by transfecting a plasmid expressing the tagged protein, followed by selection in 500 μg/ml geneticin (Life tech, cat. 10131019). After 2 weeks of geneticin selection, cells were sorted using a MoFlo flow sorter (Beckman Coulter) to ensure that they were labeled with the mEos3.2 fluorophore (excitation at 488 nm, emission at 515 nm). To test for single-molecule FRET between mEos3.2 or PA-JF$_{549}$ and JF$_{646}$ on different proteins, vectors expressing HaloTag-tagged and SNAP-tagged CENP-A were generated by inserting the HaloTag or SNAP tag sequence into the NcoI/XhoI site of the mEos3.2-tagged CENP-A vector described above[59]. The HaloTag protein alone was also expressed in the same vector as a control. The vector expressing SNAP-tagged histone H2B has previously been described[45].

**Mammalian live-cell and fixed cell single-molecule imaging**. ES cells expressing mEos3.2-HaloTag-tagged CHD4 were passaged 2 days before imaging onto 35 mm glass bottom dishes No 1.0 (MatTek Corporation P35G-1.0-14-C Case) in phenol red-free serum and mLIF conditions. Just before imaging, if necessary, cells were labeled with 5 μM HaloTag-JF$_{646}$ ligand for at least 15 min, followed by two washes in PBS and a 30-min incubation at 37 °C in media, before imaging the cells in fresh phenol red-free serum and LIF conditions containing 5 mM Trolox. In vivo fluorescence images were collected as movies of 10,000 frames at 500 ms exposure. Continuous photo-conversion was achieved using the 405 nm laser at low activation powers of ~10 W/cm$^2$.

For protein complex tracking, ES cells expressing SNAP-tagged histone H2B or mEos3.2-, HaloTag- and SNAP-tagged CENP-A were generated by transfection of the appropriate expression vectors. Four microliters of Lipofectamine® 2000 (Life tech, cat. 11668027), incubated in 100 μl of OPTI-MEM® I Reduced Serum Medium (Thermo Fisher Scientific, cat. 31985070) for 5 min, was added to approximately 2–3 μg of expression vectors, also incubated in 100 μl of OPTI-MEM® for 5 min. The mixture was then incubated further for 15 min before adding to ES cells that had been passaged at the same time onto the 35 mm glass bottom dishes. After 2 days, cells were labeled using the appropriate HaloTag ligands as described above for CHD4. SNAP tag ligands were also labeled similarly but with an initial incubation of 30 min prior to washes.

FRET was optimized by ensuring an excess of acceptor dye surrounding the donor mEos3.2 or PA-JF$_{549}$. For FRET between mEos3.2-tagged CENP-A and JF$_{646}$-tagged CENP-A, this was carried out by transfecting 0.4 μg of mEos3.2-tagged CENP-A or SNAP-tagged CENP-A alongside 2 μg of HaloTag-tagged CENP-A. For FRET between PA-JF$_{549}$-tagged and JF$_{646}$-tagged CENP-A or PA-JF$_{549}$-tagged and JF$_{646}$-tagged H2B, this was achieved by labeling cells post-transfection with 0.2 μM SNAP-tag PA-JF$_{549}$ ligand and 2 μM SNAP-tag JF$_{646}$ ligand. Finally, for CENP-A/H2B FRET, 1 μg of mEos3.2-tagged CENP-A was transfected alongside 1 μg of SNAP-tagged CENP-A and labeled with 5 μM SNAP-tag JF$_{646}$ ligand. Cells expressing the mEos3.2-CENPA, PA-JF$_{549}$-CENP-A, or PA-JF$_{549}$-H2B construct were identified by their ability to photo-activate single molecules using the 405 nm laser and cells labeled with the HaloTag-JF$_{646}$ or SNAP tag JF$_{646}$ ligands by their localization at centromeric foci or to the nucleus (for HaloTag-CENP-A or SNAP-tagged H2B, respectively), as determined by imaging using the 641 nm laser (1 kW/cm$^2$). Fixed and live cell fluorescence images were collected as movies of 3000 to 5000 frames at 500 ms exposure. Photo-conversion was achieved using 100 ms exposures of the 405 nm laser every 6 s at low activation powers of ~10 W/cm$^2$. For cell fixation, cells were washed with PBS, fixed at room temperature in PBS containing 4% formaldehyde for 15 min, washed again in PBS, and then resuspended in PBS containing 5 mM Trolox.

**Mammalian cell image processing and analysis**. Live-cell and fixed-cell single-molecule movies were analyzed using Rapidstorm software that determines single-molecule localizations from PALM movies[68], after using Image's rolling ball background correction with a radius of 5 pixels. Only fluorescent puncta less than 5 or 3 pixels wide (for Microscopes 1 and 2, respectively) and with a fixed global threshold above 25,000 were analyzed. To track single CHD4, CENP-A, or H2B molecules, we used custom code to connect single-molecule localizations and extract the length of their trajectories (script can be found at https://github.com/TheLaueLab/trajectory-analysis). Fluorescent puncta were considered to be the same molecule if they were within 100 nm between frames because we do not expect to see diffusion coefficients greater than this for bound H2B/CENP-A/CHD4. Molecules were still connected if they were not detected for 1 frame to reduce the likelihood of molecules dropping briefly below the signal-to-noise threshold. Trajectories smaller than 3 localizations are discarded to reduce the likelihood of detecting noise. The average intensity of these trajectories was also extracted for FRET efficiency calculations—we ignore the first and last frames because the molecule may not have been fluorescing throughout these frames. Single-molecule images shown in Fig. 4 were generated using Peak Fit such that localizations represent the precision at which they were localized[69]. Localization precision was calculated after the Rapidstorm analysis by fitting a histogram of nearest neighbor pairwise distances[70].

For live-cell tracking of single bound CHD4 molecules, the experiment was replicated twice on Microscope 1 and once on Microscope 2. We again chose to analyze datasets from Microscope 1 recorded on the same day to reduce systematic errors arising from subtly different microscope alignments. We collected 772 and 539 single-molecule trajectories from single-molecule movies of mEos3.2- and mEos3.2–JF$_{646}$-tagged CHD4 molecules, respectively (typically two cells were studied in each movie). Given that 10–30% of single mEos3.2–JF$_{646}$ molecules were in the on-state for longer than single mEos3.2 molecules, this is an appropriate sample size to demonstrate the change we observe.

For the CENP-A protein FRET proximity analysis, more trajectories were collected because relatively few mEos3.2-tagged molecules were expected to be next to JF$_{646}$-tagged molecules. The experiment was replicated once on Microscope 1 and twice on Microscope 2, and we analyzed one of the datasets from Microscope 2 in which we collected 17,953, 17,288, and 21,018 trajectories from single-molecule movies of mEos3.2-CENP-A, mEos3.2-CENP-A/JF$_{646}$-CENPA, and mEos3.2-CENP-A/JF$_{646}$-H2B, respectively (typically four cells were studied in each movie). Given that ~0.1–1% of mEos3.2 molecules in the presence of JF$_{646}$-tagged

molecules were in the on-state for longer than single mEos3.2 molecules, this is an appropriate sample size to demonstrate the change we observe.

For H2B FRET proximity analysis, the experiment was replicated twice on Microscope 2, and we analyzed one of the datasets in which we collected 2114 and 3970 trajectories from single-molecule movies of PA-JF$_{549}$-H2B and PA-JF$_{549}$-H2B/JF$_{646}$-H2B, respectively (typically four cells were studied in each movie). Given that 10% of PA-JF$_{549}$ molecules in the presence of JF$_{646}$-tagged molecules were in the on-state for longer than 1% of single PA-JF$_{549}$ molecules, this is an appropriate sample size to demonstrate the change we observe. Trajectories longer than 1% of single PA-JF$_{549}$ molecules were identified and colored in blue. FRET efficiency was calculated using the average intensity of molecules labeled with PA-JF$_{549}$ alone. Density maps of 2-pixels (312 nm) wide were generated to show the number of molecules found within the region and whether the average track length was below (yellow) or above 12 s (blue), a 1% confidence interval set such that there is a 1% likelihood of PA-JF$_{549}$-tagged molecules having a track length longer than 12 s.

**Data availability**. The datasets generated and analyzed during the current study are available from the authors on request.

**Code availability**. The software used for in vitro photophysical analysis and live-cell tracking can be found at https://github.com/TheLaueLab/blink-analysis and https://github.com/TheLaueLab/trajectory-analysis, respectively. Other softwares used include the open-source software micro-manager (https://www.micro-manager.org), ImageJ64,65, Rapidstorm68, and PeakFit69.

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

## Acknowledgements

We thank Tessa Kretschmann for help in preparing the figures for publication, Luke Lavis for providing the JF646 and PA-JF549 dyes, Markus Sauer for providing the SNAP-tagged histone H2B mammalian expression vector, Andy Riddell for flow cytometry, Len Packman in the Protein & Nucleic Acid Chemistry Facility (PNAC) for mass spectrometry and Pablo Hernandez-Varas for fluorescence lifetime measurements at the MRC Weatherall Institute of Molecular Medicine (Oxford). We thank Ulrike Endesfelder and Mike Heilemann for many helpful discussions regarding the concept and experimental design, and Aleks Ponjavic for comments on the manuscript. We thank the Royal Society for a University Research Fellowship to S.F.L. (UF120277) and the Wellcome Trust (206291/Z/17/Z), EC FP7 4DCellFate project (277899), and Medical Research Council (MR/P019471/1 and MR/M010082/1) for financial support.

## Author contributions

D.K. suggested the concept. S.B., L-M.N., D.L., D.K., S.F.L. and E.D.L. designed the experiments. E.J.R.T., S.B., L-M.N. and D.S. expressed, purified, and labeled the proteins. S.B., L-M.N., and D.S. performed the imaging experiments. S.B., L-M.N., E.J.R.T. and D.S. analyzed the data. K.W., W.B., L-M.N., S.B., and S.F.L. developed the software for data analysis. S.B., D.L., E.J.R.T., L.B., Y.L.T. and O.T. cloned the constructs and carried out preliminary SPT experiments. S.B., J.C. and B.H. cloned the constructs and made cell lines used for imaging. B.C.L., S.B., L-M.N. and C.E. carried out the fluorescence lifetime measurements. S.B., L-M.N., D.K., S.F.L. and E.D.L. wrote the manuscript with assistance from other authors. D.K., S.F.L. and E.D.L. supervised the project.

## Additional information

**Competing interests:** The authors declare no competing interests.

