## [Peer Review File · Nature Communications]

Reviewers' comments:

Reviewer #3 (Remarks to the Author):

The revised manuscript 'FRET enhancement of photo-modulatable fluorophores for improved single molecule tracking of proteins and complexes in live mammalian cells' by Basu et al. has addressed most of my previous concerns and the writing was significantly improved. With increased trajectory length and not-dramatically-reduced localization precision, the method could improve sptPALM by expanding the time window and including 'interaction' as a parameter. The characterization of a FRET pair between a fluorescent protein and an organic dye is also new and much needed. However, this method has several inherent limitations (see below), for which the authors should discuss thoroughly to aid readers. I also suggest a few more revisions.

1. The authors should compare this new method with single molecule tracking (SMT) using organic dye such as JF646 by itself, and discuss thoroughly the pros and cons. In short, SMT using Halo/Snap labeled with dyes can generate orders of magnitude longer trajectories and much higher localization precision compared to the donor mEos3.2 in the FRET pair; the Halo/SNAP tag by itself is also much smaller than FP-HALO/SNAP so to avoid disruption of fusion's functionality. Why not use the dye directly for SMT? The FRET method may be better suited for monitoring protein-protein interactions instead of SMT.

2. This method fundamentally relies on a trade-off between resolution and trajectory length, which is the same for conventional SMT. The improved photon budget of mEos3 relies on the FRET efficiency, which naturally sets an upper bound of how many photons the donor could emit, since higher FRET efficiency reduces the photons number per frame.

3. The authors argued that acceptor fluorescence cannot be monitored due to background fluorescence from direct excitation and conversion. As such, individual FRET states (representing different complex states) are hard to monitor. Although the authors claimed the method can be 'used to study how dynamics or residence times of specific protein complexes differ from those of the isolated protein, or other complexes containing the same protein' (line385-386), the authors only showed some events of intensity change in a single trajectory (Figure5b). They didn't calculate the FRET efficiency as Figure S10 or perform further kinetic analysis of those states. The FRET measurement by dividing the intensity of donor by the mean intensity in the absence of acceptor is inaccurate. The authors should also comment on the error of their measurements in Figure S10.

4. The authors commented that 'because the photon budget of mEos3.2 is greatly increased, this approach may be useful to improve fixed cell super-resolution imaging studies' (Line 389). As one reviewer mentioned, lower intensity in each frame and longer total on-time is not good for quantitative super-resolution imaging. The authors already specified that the localization precision of single molecule does decrease from 15nm to 28nm (line 208). More ON frames will not only increase the total imaging time, but also exhibit as self-clusters, which have to be taken into account explicitly when reconstructing the superresolution image. Therefore, the high photon budget does NOT necessarily produce better super resolution imaging.

5. A better negative control in Line 241 should be JF646 with free HALO tag expressed in the cell.

Response to the Reviewers:

Reviewer #3:

1. The authors should compare this new method with single molecule tracking (SMT) using organic dye such as JF646 by itself, and discuss thoroughly the pros and cons. In short, SMT using Halo/Snap labeled with dyes can generate orders of magnitude longer trajectories and much higher localization precision compared to the donor mEos3.2 in the FRET pair; the Halo/SNAP tag by itself is also much smaller than FP-HAL/SNAP so to avoid disruption of fusion's functionality. Why not use the dye directly for SMT? The FRET method may be better suited for monitoring protein-protein interactions instead of SMT.

We have already directly compared single-molecule tracking of one of the dyes (PA-JF₅₄₉) with one of our FRET cassettes (PA-JF₅₄₉-JF₆₄₆) and have shown that the on-state time is significantly increased using the FRET cassette (1.8 +/- 0.3 s to 4.1 +/- 0.6 s, respectively, see Supplementary Figure 9). The data in Supplementary Figures 6 and 9 also allow a comparison between the on-state times for mEos3.2-JF₆₄₆ and PA-JF₅₄₉ (13.0 +/- 0.5 s and 1.8 +/- 0.3 s, respectively). mEos3.2-JF₆₄₆ also emits 5.4×10^4 photons/switching event whereas PA-JF₅₄₉ emits 3.6×10^4 photons/switching event. We are therefore confused as to why the reviewer believes dyes can generate orders of magnitude longer trajectories – our data strongly suggests the opposite.

To address the reviewer's concerns, we have now added more discussion of when to use a dye attached to a Halo-/SNAP-Tag on its own versus one of our constructs. As the reviewer suggests, the localisation precision is higher for PA-JF₅₄₉ alone when compared to our FRET cassettes and there will always be a trade off between track length and precision. We have added this to the discussion. Of course precision is not always critical when recording long trajectories, e.g. of bound CHD4, and so we have tried to make it clearer when precision is important for sptPALM and when it is not so relevant.

A major aim of our approach was to enable tracking of single protein complexes, which is obviously not possible using one of the dyes on its own, and so we agree with the reviewer that the method may be most useful for this application. In these applications one will always need to tag both proteins, and the worries about disrupting functionality will be the same.

2. This method fundamentally relies on a trade-off between resolution and trajectory length, which is the same for conventional SMT. The improved photon budget of mEos3 relies on the FRET efficiency, which naturally sets an upper bound of how many photons the donor could emit, since higher FRET efficiency reduces the photons number per frame.

We agree with the reviewer that *'This method fundamentally relies on a trade-off between resolution and trajectory length, which is the same for conventional SMT'*, although the trade off is between localisation precision (not resolution) and track length, and so we have added this sentence directly to the discussion. However, a 2-fold decrease in intensity ($\sqrt{2}$ loss in precision) would conventionally result in a 2-fold increase in the trajectory length. Due to FRET, we are seeing mean increases in track length of 7-fold (for mEos3.2-JF₆₄₆ vs mEos3.2) as a result of a change in the photobleaching kinetics. We have now tried to make this clearer in the manuscript.

The second sentence delves into the upper bound for how many photons the donor can emit, and thus the mechanism for how FRET increases the photons emitted by the donor. The reviewer is correct that *'higher FRET efficiency reduces the photons number per frame'* but it does not necessarily set an upper

bound for how many photons the donor can emit. We hypothesise that FRET leads to a competing kinetic pathway between photobleaching and energy transfer and that it is more likely that the acceptor photobleaching rate sets the upper bound for the total number of photons a donor can emit. This would explain why mEos3.2-JF₆₄₆ and PA-JF₅₄₉-JF₆₄₆ both emit a similar total number of photons.

3. The authors argued that acceptor fluorescence cannot be monitored due to background fluorescence from direct excitation and conversion. As such, individual FRET states (representing different complex states) are hard to monitor. Although the authors claimed the method can be ‘used to study how dynamics or residence times of specific protein complexes differ from those of the isolated protein, or other complexes containing the same protein’ (line385-386), the authors only showed some events of intensity change in a single trajectory (Figure5b). They didn’t calculate the FRET efficiency as Figure S10 or perform further kinetic analysis of those states. The FRET measurement by dividing the intensity of donor by the mean intensity in the absence of acceptor is inaccurate. The authors should also comment on the error of their measurements in Figure S10.

In our work thus far, and in this paper, our goal was to develop the method and demonstrate that the approach works. In Figure 5b we indeed illustrate intensity changes in just a single trajectory, but clearly it is now possible to study many such trajectories and we are actively using the approach to study CHD4 and NuRD assembly/function. We plan to publish a detailed study of this in the future, but it will definitely require a dedicated paper.

We agree that our measurement of FRET efficiencies is inaccurate and we would be happy to remove this from the paper. To measure FRET efficiencies accurately requires simultaneous acquisition of both the donor and acceptor fluorescence. Unfortunately due to the experimental complexity we have not been able to achieve this in the time constraints of a review. We are, however, confident that in the longer term it will be possible to simultaneously detect both donor and acceptor fluorescence, and thus to calculate FRET efficiencies more rigorously, but we do not believe this is key to the paper. Here we simply wanted to use FRET to identify whether a protein is part of a complex (or not) and then to track single protein complex molecules.

For example, we have used our longer trajectories of H2B/H2B interactions to show that they exhibit periods of constrained motion interrupted by periods of linear motion. The reviewer also mentions our comment in the discussion that the approach can be used to “*study how dynamics or residence times of specific protein complexes differ from those of the isolated protein, or other complexes containing the same protein*”. To explain how this is possible, we have tried to make it clearer in the discussion. However, for example, we can use our longer trajectories to track PA-JF₅₄₉-H2B/JF₆₄₆-CENP-A and PA-JF₅₄₉-H2B/JF₆₄₆-H2B containing nucleosomes in heterochromatin. We can then compare their dynamics in mouse embryonic stem (ES) cells by plotting velocity histograms of these molecules to compare their dynamics. If we do this, we can show that H2B/CENP-A nucleosomes have a larger range of motion than canonical nucleosomes in heterochromatin (see Response Figure 1, below).

Figure 1. Comparison of instantaneous velocity histograms of PA-JF₅₄₉-H2B/JF₆₄₆-CENP-A and PA-JF₅₄₉-H2B/JF₆₄₆-H2B containing nucleosomes in heterochromatin. In mouse embryonic stem (ES) cells H2B/CENP-A nucleosomes have a larger range of motion than canonical nucleosomes in heterochromatin.

4. The authors commented that ‘because the photon budget of mEos3.2 is greatly increased, this approach may be useful to improve fixed cell super-resolution imaging studies’ (Line 389). As one reviewer mentioned, lower intensity in each frame and longer total on-time is not good for quantitative super-resolution imaging. The authors already specified that the localization precision of single molecule does decrease from 15nm to 28nm (line 208). More ON frames will not only increase the total imaging time, but also exhibit as self-clusters, which have to be taken into account explicitly when reconstructing the superresolution image. Therefore, the high photon budget does NOT necessarily produce better super resolution imaging.

After further consideration, we agree with the referee that our approach may not necessarily improve existing super-resolution imaging approaches and so we have now removed this comment from the discussion. Our major advantage is the ability to track proteins and protein complexes for long periods of time.

5. A better negative control in Line 241 should be JF646 with free HALO tag expressed in the cell.

We have now carried out this experiment as requested (see Response Figure 2, below), and we have added this control experiment to the paper in Supplementary Figure 7. There is a slight increase in the presence of highly expressed free HaloTag protein, but not as much as when a protein interaction occurs.

Figure 2. (a) Representative 500 ms exposure images from the middle of the nucleus of mEos3.2-tagged CENP-A alone and with JF₆₄₆-tagged HaloTag (left). Images of the JF₆₄₆ dye show that the HaloTag has been successfully labeled (right). (b) Montages of three representative 500 ms exposure single-molecule traces (one image for every 4 frames) are shown in the absence or presence of the JF₆₄₆-labeled HaloTag. (c) Histograms showing the percentage of molecules remaining with a particular ‘on-state’ time (*i.e.* individual track length) after photo-conversion when performing sptPALM in mouse ES cells. Cells expressing mEos3.2-tagged CENP-A are compared with cells also expressing JF₆₄₆-tagged HaloTag.

REVIEWERS' COMMENTS:

Reviewer #3 (Remarks to the Author):

The revised manuscript adequately addressed my concerns, and I especially applaud the authors' efforts in clarifying claims with specifics and conducting new control experiments. I hereby recommend its acceptance for publication.